# Glucocorticoid-Induced Leucine Zipper Protein and Yeast-Extracted Compound Alleviate Colitis and Reduce Fungal Dysbiosis

**DOI:** 10.3390/biom14101321

**Published:** 2024-10-17

**Authors:** Marco Gentili, Samuele Sabbatini, Emilia Nunzi, Eleonora Lusenti, Luigi Cari, Antonella Mencacci, Nathalie Ballet, Graziella Migliorati, Carlo Riccardi, Simona Ronchetti, Claudia Monari

**Affiliations:** 1Department of Medicine and Surgery, Pharmacology Division, University of Perugia, 06132 Perugia, Italyeleo95@gmail.com (E.L.); luigi.cari@unipg.it (L.C.); graziella.migliorati@unipg.it (G.M.); carlo.riccardi@unipg.it (C.R.); 2Department of Medicine and Surgery, Medical Microbiology Division, University of Perugia, 06132 Perugia, Italy; samuele.sabbatini@unipg.it (S.S.); antonella.mencacci@unipg.it (A.M.); 3Department of Medicine and Surgery, University of Perugia, 06132 Perugia, Italy; emilia.nunzi@unipg.it; 4Lesaffre Institute of Science & Technology, Lesaffre International, 59700 Marcq-en-Baroeul, France; n.ballet@lesaffre.com

**Keywords:** *Candida*, *Meyerozyma*, colitis, glucocorticoid-induced leucine zipper, mycobiome, prebiotic

## Abstract

Inflammatory bowel diseases (IBD) have a complex, poorly understood pathogenesis and lack long-lasting effective treatments. Recent research suggests that intestinal fungal dysbiosis may play a role in IBD development. This study investigates the effects of the glucocorticoid-induced leucine zipper protein (GILZp)”, known for its protective role in gut mucosa, and a yeast extract (Py) with prebiotic properties, either alone or combined, in DSS-induced colitis. Both treatments alleviated symptoms via overlapping or distinct mechanisms. In particular, they reduced the transcription levels of pro-inflammatory cytokines IL-1β and TNF-α, as well as the expression of the tight junction protein Claudin-2. Additionally, GILZp increased MUC2 transcription, while Py reduced IL-12p40 and IL-6 levels. Notably, both treatments were effective in restoring the intestinal burden of clinically important *Candida* and related species. Intestinal mycobiome analysis revealed that they were able to reduce colitis-associated fungal dysbiosis, and this effect was mainly the result of a decreased abundance of the *Meyerozima* genus, which was dominant in colitic mice. Overall, our results suggest that combined treatment regimens with GILZp and Py could represent a new strategy for the treatment of IBD by targeting multiple mechanisms, including the fungal dysbiosis.

## 1. Introduction

Inflammatory bowel diseases (IBD) encompass two important chronic pathologies, namely Crohn’s disease and ulcerative colitis. Both diseases are characterized by abdominal pain, diarrhea, weight loss, bloody stools, and an inflammatory state owing to granulocytic infiltration in the lamina propria, and this, in turn, causes the persistent release of pro-inflammatory cytokines [1,2]. IBD is also characterized by cycles of alternating relapse and remission phases, so they impact and reduce the quality of life of patients. Although the pathogenesis of Crohn’s Disease and ulcerative colitis has not been completely elucidated, it is known that genomic predisposition, environmental stimuli, dysbiosis of the intestinal microbiota, and aberrant immune responses play pivotal roles in their development. So far, no pharmacological treatment is available for the definitive cure of IBD, and the existing options are not suitable for all patients. The most common strategy is to use anti-inflammatory drugs or biological agents against inflammatory targets with potentially significant adverse effects [3]. The aim of these treatments is to achieve and prolong clinical remissions.

Ongoing research into new pharmacological approaches has resulted in the use of recombinant proteins in experimental mouse models of (IBD) [4,5]. In particular, glucocorticoid-induced leucine zipper (GILZ)-based recombinant proteins present a potentially efficacious treatment for IBD because of their glucocorticoid-like anti-inflammatory properties [6,7,8]. Additionally, we have recently demonstrated that an exogenously administered GILZ protein can control the permeability of the gut in a mouse model of colitis [9], and that GILZ may act as a secretory protein, being expressed by goblet cells in human specimens [10]. Therefore, the potential therapeutic effect of GILZ-based proteins in IBD involves both the regulation of the integrity and permeability of the gut and the maintenance of the homeostasis of immune cells.

Over the last few years, numerous studies have documented that patients with IBD exhibit significant changes in the composition and functionality of the gut microbial communities that result in an imbalance between protective and harmful bacteria [11,12,13]. The exact mechanisms by which this intestinal dysbiosis contributes to IBD have not been yet fully understood, but it is believed that alterations in the microbial communities can affect intestinal barrier function, mucosal immune responses, and the production of both metabolites and inflammatory molecules. Until a few years ago, the majority of studies focused on the involvement of the bacterial component of the intestinal microbial communities. Only recently, advancements in sequencing technologies and bioinformatics tools have led to significant progress in the understanding of the composition and function of the intestinal fungal community, as well as the association between fungal dysbiosis and intestinal inflammation [14,15,16,17,18,19,20]. Indeed, it has been reported that changes in fungal diversity and abundance can contribute to the development or exacerbation of intestinal inflammation in IBD [17,18,21,22,23,24,25,26,27]. Although a specific fungal pattern that can define the intestinal mycobiota in patients with IBD has not yet been identified, several studies have reported an increase in the populations of intestinal *Candida* spp., as well as former *Candida* spp., including *Debaryomyces hansenii* (ex *C. famata)*, *Nakaseomyces glabrata* (ex *C. glabrata)*, *Kluyveromyces marxianus* (ex *C. kefyr*), *Pichia kudriavzevii* (ex *C. krusei*), *Meyerozyma guillermodii* (ex *C. guilliermondii*), and *Clavispora lusitaniae* (ex *C. lusitaniae*), both in human patients and in preclinical models of IBD [21,28,29,30,31,32,33,34,35,36,37]. These recent findings suggest that future therapeutic interventions targeting fungal dysbiosis could be a promising line of research for IBD management.

Prebiotics are emerging as promising new treatment strategy for IBD [38,39]. Indeed, there is substantial evidence to suggest their potential benefits through multiple mechanisms of action, including promotion of the growth of beneficial bacteria, such as Bifidobacteria and Lactobacilli, increased production of short-chain fatty acids (SCFA), maintenance of gut barrier integrity, regulation of immune responses, and symptom relief [40,41,42,43,44]. Currently, the range of confirmed prebiotic substances is limited, with galactans and fructans (e.g., inulin) dominating the market. However, the need to target a wide spectrum of commensal organisms has led to the exploration of new prebiotic candidates, including plant-derived carbohydrates, animal-derived substrates, non-carbohydrate compounds, and fungal-based substances [45]. Among these, extracts from *Hericium erinaceus* (*H. erinaceus*), a Chinese-origin fungus, has shown great promise as a potential prebiotic for managing IBD [46]. Beneficial effects of *H. erinaceus* have been demonstrated in experimental models of IBD [47,48,49,50,51], in one ex vivo study [52], and in healthy volunteers [51]. Recently, a multicentric retrospective study suggested a real-world therapeutic potential combination of *H. erinaceus* with quercetin, berberine, niacin, and biotin in improving 5-ASA’s ability to induce remission in mild-to-moderate ulcerative colitis [53].

While most of the research on prebiotics has historically focused on their impact on bacterial communities in the gut, to the best of our knowledge, no study has specifically investigated the impact of prebiotics on the IBD-associated fungal community.

In light of the potential advantages of prebiotics in patients with IBD, it is important to provide data about their role in the treatment of IBD, especially in combination with molecules that exhibit pharmacological activity. 

Given these considerations, the objective of our study was to evaluate the potential therapeutic effects of a recombinant GILZ protein (GILZp) and a yeast extract compound (Py), a potential prebiotic, both individually and in combination, in a murine model of colitis. Our results demonstrate that this treatment combination effectively attenuated dextran sulfate sodium (DSS)-induced colitis by reducing inflammation, restoring epithelial barrier integrity, and ameliorating fungal dysbiosis by re-establishing the abundance of the genus *Meyerozyma*.

## 2. Materials and Methods

### 2.1. Animals

Male C57BL/6 mice (6–8 weeks old) were purchased from ENVIGO (Udine, Italy). The mice were housed under specific pathogen-free conditions and a 12/12 h light/dark cycle and received water and food *ad libitum*. The animal care and experimental procedures have been approved by a research ethics committee at the institution at which the studies were conducted, using protocols approved by the Italian Ministry of Health (approval no. 754/2020-PR). The animal studies are reported according to the ARRIVE guidelines [54]. Mice were randomly allocated to five groups of five mice each: one non-colitic and four colitic groups. Colitis was induced by adding 3% DSS (36–55 KDa, TdB Consultancy AB, Uppsala, Sweden) to autoclaved drinking water ad libitum for 5 days. From day 0 to day 8, the four colitic groups were subjected to the following treatments: PBS (200 μL, control group), GILZp (0.2 mg/kg) (consisting of the whole GILZ protein with a cell-permeation fused peptide, also known as TAT-GILZ) (1), Py (1000 mg/kg), and Py + GILZp. Py dose (1000 mg/kg) was chosen after testing 100, 500, 1000 mg/Kg in a preliminary experiment (Appendix A). GILZp and PBS were intraperitoneally administered whereas Py was orally administered by gavage, since it was considered an orally available potential prebiotic. All of the mice were fed standard chow for the entire experiment. DAI and weight were evaluated daily in each mouse (Table 1) [9]. 

On day 10, the mice were sacrificed, and their colons were removed, emptied, weighted, and processed to undergo specific analyses.

### 2.2. Compounds

The candidate prebiotic used in this study (NuCel^®^ 582 MG) is a primary yeast (Py) extract obtained by using a selected strain of *Saccharomyces cerevisiae* grown on a molasses-based media. Py is a protein hydrolysate obtained by the plasmolysis of high-protein yeasts cells and autolysis of yeast proteins. The process involves breaking down proteins into polypeptides and free amino acids. Cell walls and soluble fractions are separated through centrifugation. Soluble fractions are concentrated and dried to obtain the final micro-granulated Py. It can promote the growth of lactobacilli and is also suitable for ripening flora. This product was provided by Procelys (Lesaffre) (Table 2 and Table 3).

### 2.3. RNA Extraction

mRNA was extracted from the colons using theRNeasy^®^ Plus Mini Kit (Qiagen, Hilden, Germany), following the manufacturer’s instructions. Briefly, after the addition of RLT buffer and β-mercaptoethanol, the samples were homogenized and subsequently transferred into the “gDNAEliminator spin column”. After the addition of 70% ethanol, RNA was recovered with a RNAeasy Mini spin column. The extracted mRNA was purified by contaminating DNA with the wipeout buffer. Conversion of total mRNA to cDNA was performed using a High-Capacity cDNA Reverse Transcription Kit (Qiagen, Hilden, Germany).

### 2.4. Real-Time Quantitative PCR 

Real-time quantitative PCR (RT-qPCR) was performed using the 7300 Real Time PCR System (Applied Biosystem, Waltham, MA, USA) and cDNA was amplified by using TaqMan Gene Expression Master Mix (Applied Biosystem). The amplification was performed using the QuantStudio™ 1 RT-qPCR System (Applied Biosystem) with the TaqMan Gene Expression Master Mix (Applied Biosystem). The following probes for the selected genes were used: IL-1β, TNFα, IL-6, IL-12p40, MUC2, Claudin-2, and 18S (housekeeping gene) (Thermo fisher, Waltham, MA, USA). For data analysis, the relative expression levels were calculated using the 2^−ΔCt^ method.

### 2.5. Validation of Primer Sequences for Candida spp. and Former Candida spp.

Previously published [38] forward primer (5′-GGATCTCTTGGTTCTCGCATC-3′), reverse primer (5′-AACGACGCTCAAACAGGCAT-3′) and 5′ FAM GEN_probe (5′-CGCAATGTGCGTTCAA-3′) were chosen for RT-qPCR assay. To check for the specificity of the chosen primers and probe, the nucleotide sequences of *C. albicans* (OR116194.1), *C. parapsilosis* (OR105646.1), *C. dubliniensis* (OP382361.1), *C. tropicalis* (OR105801.1), *C. sake* (OR105215.1), *Debariomyces hansenii* (ex *C. famata*) (OR105219.1), *Nakaseomyces glabrata* (ex *C. glabrata*) (OP895149.1), *Kluyveromyces marxianus* (ex *C. kefyr*) (OQ520311.1), *Pichia kudriavzevii* (ex *C. krusei*) (OR091281.1), *Meyerozyma guillermodii* (ex *C. guilliermondii*) (OR105223.1), and *Clavispora lusitaniae* (ex. *C. lusitaniae*) (OR075990.1) were retrieved from the Nucleotide database (https://www.ncbi.nlm.nih.gov/nuccore). Next, the nucleotide sequences were aligned with the primers and probe by using the CLC Sequence Viewer 7.0.2 software (QIAGEN), with a gap open cost of 10 and a gap extension cost of 1. 

### 2.6. RT-qPCR of Candida spp. and Former Candida spp.

*Candida* spp. and former *Candida* spp. were quantified using RT-qPCR. Total DNA was extracted from fresh stool samples as described below. Two sets of reactions were performed: the first one to detect the quantity of *Candida* spp. by using the 5′FAM GEN probe described above, and the second using ITS1-2 primers to amplify Fungal rDNA (2). Each sample was analyzed in triplicate, using 5 ng of DNA in a final reaction volume of 12 µL. Differences between the cycle threshold (Ct) of *Candida* spp. and ITS1-2 were calculated (ΔCt), and data were shown as 2^−ΔCt^. 

### 2.7. Mycobiome Sequencing and Analysis

Fresh stool samples were collected at day 10 of the experiment and immediately stored in dry ice. Stools were than processed for DNA extraction using an E.Z.N.A.^®^ Stool DNA Kit (Omega bio-tek) following a proper protocol provided by the kit. DNA purity and quantity was checked using a spectrophotometer and prepared for sequencing. The sequencing was outsourced to BMR Genomics (Padova, Italy). Sequencing and amplification were performed with 5 µL of DNA, and the ITS1 region of the fungal RNA gene was amplified using the following Illumina-tailed primers: ITS1F (5′-TCGTCGGCAGCGTCAGATGTGTATAAGAGACAG-CTTGGTCATTTAGAGGAAGTAA-3′) and ITS2R (5′-GTCTCGTGGGCTCGGAGATGTGTATAAGAGACAG-GCTGCGTTCTTCATCGATGC-3′). HiFi Platinum Taq (Thermo Fisher Scientific Inc., Waltham, Massachusetts, USA) was applied for PCR using the following protocol: 94 °C for 2 min; 45 cycles at 94 °C for 30 s, 55 °C for 30 s, and 68 °C for 30 s; and a final extension at 68 °C for 7 min. PCR amplicons were purified with Thermolable Exonuclease I (NEB) diluted to 1:2 and amplified according to the Nextera XT Index protocol (Illumina Inc., Sad Diego, CA, USA). The amplicons produced were normalized by means of the SequalPrep™ Normalization Plate Kit (Thermo Fisher Scientific Inc.) and multiplexed. The pool was purified with 1X Magnetic Beads Agencourt XP (Beckman Coulter Brea, CA, USA), loaded on the MiSeq System (Illumina Inc., USA), and sequenced following the V3-300PE strategy.

A total of 2980734 high-quality sequences were obtained after trimming and quality screening of raw reads obtained by Illumina MiSeq sequencing. Only reads that were at least 100 nucleotides (nt) long were retained. All of the sequenced files were subjected to a quality control procedure with the FASTQC software(0.11.9), and were then imported and analyzed using the next-generation microbiome bioinformatics platform Qiime2 (version 2022.2) implemented in our distributed cloud-computing environment (3,4,5). The forward and reverse paired-end reads were imported and analyzed using the Qiime2 2022.2 platform in a genomic cloud computing environment. At first, paired-end sequences were denoised, dereplicated, and filtered to eliminate any phiX reads and chimera (by consensus) by using the q2-dada2 quality control method for detecting and correcting (where possible) Illumina amplicon sequence data. In particular, the q2-dada2 method uses sequence error profiles to obtain putative error-free sequences, referred to as either sequence variants (SVs) or 100% operational taxonomic units. It also truncates forward and reverse sequences at the first instance of a quality score less than or equal to 2. Reads with more than five incorrect bases were discarded, and only reads with a minimum overlap of 12 nt were retained and joined. SVs were taxonomically classified using a machine learning algorithm based on the Naive Bayes classifier model trained on the full reference sequences of the UNITE Qiime formatted database, version 10.5.2021. The supervised trained classifier was then applied to the obtained SVs for mapping them according to taxonomy. A phylogenetic tree was constructed via sequence alignment with MAFFT, and the alignment was filtered and FastTree was applied to generate the tree. Analysis of the rarefaction curves of the Shannon index indicated good sequencing quality, as the richness index does not increase significantly with sampling depth for each sample Rare taxa, defined as sequence variants that were counted less than 10 times or were present in no more than three samples, were then removed from the sequenced samples. Further analyses were conducted using both the Qiime2 (ver. 2022.2) platform and R version 4.2.1 (23 June 2022) in RStudio 2022.02.3.

The within-sample alpha-diversity was assessed based on the UNITE ITS rRNA gene sequencing data, using Observed features, Chao1, and Shannon diversity indexes estimated using the QIIME2 platform. The corresponding statistical significance of the comparison of sample groups was determined using the Kruskal–Wallis and Wilcoxon pair-wise tests with FDR adjustment. Beta-diversity was calculated for the SVs using Jaccard and Bray–Curtis distances estimated in QIIME2. To reduce the dimensionality of the diversity investigation, a principal coordinate analysis (PCoA) was performed using the resulting distance matrices in order to quantitatively separate all sources contributing to the beta diversity. The first two components were used for visualization of the most effective relationships contributing to diversity between groups of samples. Pairwise comparisons were assessed with a non-parametric test (BH-adjusted *p*-value). LEfSe was used to test the association at each taxonomic level. LEfSe employs a non-parametric Kruskal–Wallis sum-rank test to differentiate between class features and a subsequent linear discriminant analysis (LDA) to estimate the effect size of taxa that violated the null hypothesis. LEfSe was applied with default alpha values for the ANOVA and Wilcoxon test (0.05), and the LDA effect size was evaluated by setting the absolute value of the logarithmic LDA threshold to 2.0, with the default LEfSe parameters. Mycobiome core analyses were evaluated with the R software, version 4.2.1.

### 2.8. Statistical Analyses

All data are expressed as mean ± standard error. Differences between mean values were tested using the GraphPad Prism 6.0 software. Statistical analyses were performed using one-way analysis of variance (ANOVA) with Tukey’s *post hoc* test. Statistical significance was indicated as *^,#^
*p* < 0.05, **^,##^
*p* < 0.01, ***^,###^
*p* < 0.001. 

## 3. Results

### 3.1. Amelioration of DSS-Induced Colitis by Treatment with GILZp and Py

We previously demonstrated that treatment with a recombinant GILZ protein (GILZp) can alleviate symptoms of DSS-induced colitis by restoring gut permeability [9]. This is crucial because a leaky gut epithelium allows for microbial communities to translocate across intestinal cells, triggering inflammation as the main cause of IBD. Despite the focus on restoring gut balance (eubiosis) in IBD with probiotics and prebiotics, limited or no beneficial effects have been achieved with prebiotics as adjuvant treatment [55,56,57]. Therefore, we investigated Py, a potential new yeast-extracted prebiotic, combined with GILZp in a mouse model of colitis. Py was administered in escalating doses daily via gavage for 8 days, starting from day 0, with administration ceasing for the final 2 days. Despite the susceptibility of these mice to DSS-induced mortality [58], those treated with Py showed improved survival rates compared to DSS control (Ctrl) (Appendix A). Survival was dose-dependent, with the highest dose (1000 mg/kg) ensuring survival until sacrifice, and thus being utilized in subsequent experiments.

To investigate the effect of combined treatment with GILZp and Py, mice were randomly allocated into five groups, including one group that comprised non-colitic mice and four groups that comprised mice with DSS-induced colitis. Among the colitic groups, one group was left untreated (Ctrl), one group was treated with only Py, one group was treated with only GILZp, and one group received the combined GILZp–Py treatment. The treatments were administered daily until day 8 (Figure 1A). During the 10-day follow-up period, we observed that mortality was reduced with both the singular and combined treatments, whereas mortality was recorded from day 8 for the DSS Ctrl group (Figure 1B). Body weight tended to decrease in all the colitic groups, with no statistically significant difference between the treatment and Ctrl group (Figure 1C). In contrast, the disease-associated index (DAI) score was significantly different across groups; in particular, the only Py and combined GILZp–Py treatment groups had a lower DAI score than the Ctrl group (Figure 1D). However, on the day of sacrifice (day 10), no differences were observed in either bodyweight or DAI score. Notably, all treatments resulted in a significant reduction in the colon weight/length ratio in comparison with the Ctrl group, suggesting mild inflammation in the treated groups (Figure 1E).

### 3.2. GILZp and Py Contribute to Reduce Colon Inflammation

Since one of the main and more striking observations was the reduction or even absence of mortality when Py and GILZp were used for treatment, we tried to understand the molecular mechanisms underlying this effect. To this end, we analyzed the main pro-inflammatory cytokines associated with DSS-induced colitis, such TNFα, IL-1β, IL-12p40, and IL-6, at the transcription level. The results show that Py was able to suppress the expression of all the inflammatory cytokines, whereas GILZp was able to suppress IL-1β and TNFα expression (Figure 2A). Furthermore, with the combined treatment, all the pro-inflammatory cytokines, except for IL-12p40, were downregulated; this effect is probably attributable to Py (Figure 2A,B). 

In the pathogenesis of colitis, the increase in the permeability of the intestinal mucosal barrier and loss of mucosal function are detrimental factors that need to be corrected for mucosal healing. We previously demonstrated that GILZp can increase ZO-1 expression [9]; therefore, we here analyzed the expression of Claudin-2, one of the most important tight junction molecules, whose expression is upregulated in the inflamed gut during colitis and is associated with an increase in the permeability of the epithelial layer [59]. We observed a significant downregulation of Claudin-2 expression in treatment with only GILZp or only Py, as well as in combined treatment with both (Figure 2C). Another key protein involved in mucosal barrier protection is MUC2, the primary component of the intestinal mucus layer, which is typically depleted in IBD [60,61,62,63]. GILZp, whether administered alone or in combination, stimulated MUC2 transcription, unlike Py alone, which did not produce this effect (Figure 2C).

### 3.3. Restoration of the Intestinal Burden of Clinically Important Candida and Former Candida Species Using GILZp and Py

Recent research has provided evidence for the involvement of the intestinal fungal community, particularly *Candida* spp., in the pathogenesis of IBD [21,30,33,64]. To investigate the effects of treatment with Py, GILZp, or combination of both (Py + GILZp) on the intestinal burden of clinically relevant *Candida* and former *Candida* spp., we conducted an RT-qPCR analysis using a previously established protocol [65]. First, we validated the assay’s proficiency in identifying our target species, including *C. albicans*, *C. parapsilosis*, *C. dubliniensis*, *C. tropicalis*, *C. sake*, *Debariomyces hansenii* (ex *C. famata*), *Nakaseomyces glabrata* (*ex C. glabrata*), *Kluyveromyces marxianus (ex C. kefyr*), *Pichia kudriavzevii* (*ex C. krusei*), *Meyerozyma guillermodii* (*ex C. guilliermondii*), and *Clavispora lusitaniae* (ex. *C. lusitaniae*) (Appendix A) [66]. The alignment confirmed the specificity with the target sequences, so we proceeded with the RT-qPCR. Our results show that the administration of DSS significantly increased the overall intestinal burden of these fungal species in Ctrl compared to the non-colitic group. Importantly, both individual treatments and their combination led to a significant reduction in the same fungal species, and their levels were similar to those observed in non-colitic mice (Figure 3).

### 3.4. Reduction in Intestinal Fungal Dysbiosis in Colitic Mice Using GILZp and Py

Next, we performed a mycobiome analysis using high-throughput internal transcribed spacer 1 sequencing of fungal ribosomal DNA to characterize the impact of the treatments on the biodiversity of the DSS-associated gut mycobiome and determine its composition. The total number of raw sequenced reads was 4,014,891, including 1,067,582, 821,557, and 1651783 reads in the non-colitic, Crtl, and Py + GILZp/Py/GILZp groups, respectively, with corresponding average values of 213,516.4, 136,926.17, and 117,984.5. In order to facilitate the comparison of fungal composition between groups and compensate for biases induced by the sequencing library size, bioinformatics analyses were carried out using the rarefaction strategy. Rarefaction curves of the Shannon (Appendix A) index evaluated on each sample versus sequencing depth shows the goodness of the sequencing data. The curves reached a plateau, thus confirming that the sample sequencing depth was adequate for comparing both the evenness and the richness in the samples. The distributed ranges of the alpha indexes Shannon, Observed Features, and Simpson (Figure 4A) in each group evaluated on data rarefied with 4000 reads were indicative of the skewed distribution of fungal species in the colitic groups: compared to the Crtl group, on average, the non-colitic (Shannon p.adj = 0.036) and treatment groups had increased evenness, while the non-colitic group had increased richness and the treated groups had reduced richness (Observed_features p.adj = 0.284, and Simpson p.adj = 0.016). Furthermore, alpha values of the treated were, on average, close to those of the non-colitic mice, thus indicating that the treatments were able to restore evenness to a point that was close to that of the non-colitic mice, thus reducing the dysbiosis. All three treated groups showed similar median values for each alpha index, with the average richness being lower than that of the Crtl group and the evenness being comparable to that of the non-colitic group. Overall, these findings indicate that colitic mice exhibit reduced evenness and richness of their gut fungal community in comparison to non-colitic mice. Both singular and combined treatments showed a clear trend toward restoring the goodness value of alpha diversity, and this indicates their potential to improve the evenness of the fungal community in the gut of colitic mice.

A permutational multivariate analysis of variance (PERMANOVA) based on non-phylogenetic beta dissimilarity matrices (Jaccard and Bray–Curtis) revealed significant fungal biodiversity in the compositions of groups (Figure 4B). The difference in the Jaccard metric (p. adj = 0.024) between the non-colitic and control mice confirms the difference in their community structure. The qualitative (Jaccard) and quantitative (Bray–Curtis) distances of the treated groups present similar values and, thus, further association analyses are needed to evaluate any differences in their fungal composition. Furthermore, since the treated group has significantly higher values than the control group in terms of both beta diversity indices, the corresponding samples have a more diverse mycobiome both in terms of abundance values and taxa types. According to the Bray–Curtis metric, significant differences were observed between the Py + GILZp and control groups (p.adj = 0.024) as well as between the GILZp and control groups (p.adj = 0.0036). PCoA cluster analyses (Figure 4C) of beta diversities confirmed the significant compositional diversity of the mycobiome among the different groups. Indeed, the first component of the Jaccard index indicated significant differences in intestinal fungal communities between the non-colitic and control or treated groups. This indicates that the non-colitic mouse intestine is characterized by specific taxa that are not present in other groups, i.e., a higher degree of richness. Moreover, the first component of the Bray–Curtis metric underlines significant differences in the proportion of fungi between treated mice and non-colitic or control mice, thus emphasizing the presence of characteristic distributions of taxonomies within each group. The control group, therefore, presented with significant differences both in the number of types of fungal taxa and in their proportions in comparison with each of the other groups. 

Overall, these results indicate that the fungal communities of colitic mice are characterized by richness values that are lower than those in non-colitic mice and that the fungal taxa found in the treated groups are probably also present in the colitic group, but with significant differences in their abundance. 

To investigate the composition of the intestinal mycobiota in response to treatment with one compound or combined treatment, taxonomic analyses were performed at various levels, ranging from the phylum to the species level. Our results reveal that the intestinal mycobiota consisted of fungi belonging to the Ascomycota and Basidiomycota phyla across all groups, with Ascomycota phylum being the dominant one (Figure 5A).

At the genus level, the heatmap (Figure 5B) and bar plot of the taxonomies (Appendix A) show that the most abundant (on average) five genera were *Meyerozyma, Rhodotorula*, *Alternaria*, *Fusarium*, *Aspergillus*, *Mycosphaerella*, *Vyshiacozyma*, *Cladosporium*, *Filobasidium*, and *Sporobolomyces*. Of note, *Meyerozyma* was the dominant genus in the control group, and its abundance was greatly reduced by singular and combined treatments. Because long-lived microbes in a community modulate the physiology of the host microbial system, and given the variable and inherently compositional nature of metagenomic data, we focused our further analyses on the most abundant and prevalent taxa. Thus, to investigate dysbiosis and differential abundance as key determinants of colitics and non-colitics, we identified the core microbial group, defining it as bacteria present at ≥0.1% relative abundance in ≥50% of all samples (Figure 5C) [67,68,69]. The following genera showed 100% prevalence in their respective groups (minimum abundance is reported in parentheses): *Aspergillus* (10%), *Mycosphaerella* (5%), *Fusarium* (5%), *Alternaria* (5%), *Visnhiacozyma* (4%), *Cladosporium* (3%), and *Trichoderma* (1%) in the non-colitic group; *Meyerozyma* (20%), *Fusarium* (1%), *Alternaria*, (1%), *Aspergillus* (1%), *Sporobolomyces* (1%), *Cladosporium* (1%), and *Visnhiacozyma* (1%) in the control group; *Fusarium* and *Mycosphaerella* (5%), *Spencerozyma* (2%), *Alternaria* (2%), *Aspergillus* (2%)*, Meyerozyma* (2%), and *Candida* (1%) in the Py group; *Alternaria* (4%) and *Aspergillus* (1%) in the GILZp group; and *Cladosporium* (4.5%), *Mycosphaerella* (2.5%), *Fusarium* (1%), *Aspergillus* (1%), and *Microdochium* (1%) in the Py + GILZp group. To identify the fungal taxa driving these community shifts, a linear discriminant analysis effect size (LEfSe) analysis was subsequently performed from the phylum to the species level. Figure 6 shows the results of the LEfSe analysis conducted at the genus level through pairwise comparisons of the control group with the non-colitic (a), Py (b), GILZp (c), or Py + GILZp (d) groups. Among the core genera with a prevalence of 100%, highlighted with black arrows, the LEfSe analysis demonstrated a consistent and significant association of the *Meyerozyma* genus with the colitic group in all the comparisons conducted (Figure 6A–D). On the other hand, several genera, such as *Aspergillus*, *Mycosphaerella*, *Fusarium*, *Vishniacozyma*, and *Trichoderma*, showed significant associations with the non-colitic group (Figure 6A). Our findings indicated that Py and Py + GILZp groups, compared to the control mice, were enriched with *Mycosphaerella* (Figure 6B,D). None of the genera present in the GILZp mycobiome core were significantly associated with the GILZp group (Figure 6C). Further LEfSe analyses of the same pairwise groups at the species level showed that *M. guillermondii* and *M. tassiana* were significantly associated with the control group and the non-colitic or Py + GILZp group, respectively. These results suggest that the *Meyerozyma* genus could be one of the key microbial genera characterizing the gut fungal community in DSS-induced colitis. Furthermore, the reduction in its abundance may contribute to the shift of the intestinal fungal community toward eubiosis, thereby contributing to the observed anti-colitic effects with both singular and combination treatments.

## 4. Discussion

Despite the availability of several classes of drugs for the relief of symptoms and reduction in inflammation in IBD, many patients either did not respond to treatment or experienced a loss of response over time and, often, needed to switch to another class of drugs. However, to date, no definitive cure has yet been found for IBD. Additionally, long-term treatments with some efficacious drugs, such as glucocorticoids and other immunosuppressants, lead to serious adverse effects, including metabolic alterations and increased risk of infections and malignancies [70,71]. GILZ is a protein that mediates several anti-inflammatory effects of glucocorticoids, and previous studies on GILZp have reported promising results for the treatment of IBD [6,8,72,73,74,75]. Considering that GILZ is also involved in the pathophysiology of IBD, as demonstrated in both preclinical models of colitis and in patients, in this study, we investigated the effects of combining GILZp with a potential prebiotic yeast extract (Py) in a mouse model of colitis. In our initial experimental model of colitis, the mortality observed in the control mice was considerably prevented by Py in a dose-dependent manner. Based on this finding, one hypothesis that explains the observed effect is that Py could play a protective role in the mucosa to mechanically prevent the disruption of the mucosal barrier, but further studies are needed to demonstrate this hypothesis. However, the treated mice in this study showed symptoms of IBD, such as loss of bodyweight and a DAI score that was similar to that of control mice at the end of the experiment. Alternatively, similarly to prebiotics, Py may favor the growth of beneficial microbiota, and this may ultimately relieve the symptoms of colitis. Interestingly, treatment with only GILZp led to the survival of colitic mice, with the molecular events underlying this beneficial effect only partially overlapping with those of Py. These effects were expected, since the treatment with the whole protein of GILZ previously lead to amelioration of symptoms via several mechanisms, including anti-inflammatory and barrier-protective effects [9,76,77]. In our experimental conditions, GILZp, by dampening IL-1α and TNFβ levels and increasing MUC2 transcription, improved the symptoms of colitis, thus helping resolve inflammation and probably stimulating mucus production. On the other hand, Py acted exclusively on all pro-inflammatory cytokines by either inhibiting their production or reducing their expression. These findings are in line with previous in vitro and in vivo studies on both traditional and potential prebiotics (for example extracts from *H. erinaceus*) which exert an anti-inflammatory effect in IBD via multiple mechanisms, including reduction in pro-inflammatory cytokines [45,52,78]. Of note, both GILZp and Py reduced Claudin-2 mRNA; this implies that they both can play a role in improving barrier integrity [79]. Overall, both GILZp and Py treatments promoted the survival of mice and led to milder disease by acting on both distinct and overlapping molecular targets. Although no synergistic action was observed, the effect on separate targets contributed to the relief of symptoms. One limit of this study is the short follow-up time, since a longer experimental duration can lead to potential differences between treated and control groups in the final DAI score.

Recent studies have suggested that intestinal fungal dysbiosis potentially contributes to the development and persistence of IBD [21,22,33,35,64,79,80,81]. We therefore aimed at examining the impact of administering Py, GILZp, and a combination of both on the composition of the intestinal fungal community. Initially, we conducted a RT-qPCR analysis of fecal samples obtained from non-colitic, colitic, and treated colitic mice to determine the intestinal load of clinically important *Candida* and former *Candida* spp., namely, *Debaryomyces hansenii*, *Meyerozyma guillermondii*, and *Clavispora lusitaniae*, previously known as *Candida famata, Candida guillermondii*, and *Candida lusitaniae*, respectively [66]. In accordance with previous findings [35,82,83], our results revealed a significant increase in their overall intestinal load in colitic mice compared to the non-colitic mice. The administration of Py, GILZp, or their combination restored their intestinal load and brought it up to levels similar to those observed in non-colitic mice. We cannot exclude the direct effect of Py or GILZp on *Candida* spp., but it is plausible that this outcome is the result of several indirect effects that are primarily driven by stimulation of the growth of beneficial intestinal bacteria. Indeed, as a prebiotic [39], Py can induce the selective proliferation of bacteria, such as *Lactobacillus* and *Bifidobacterium* species [84]. Both species are well known for their capacity to reduce the overgrowth of opportunistic pathogens, such as *Candida* spp., by multiple mechanisms, including competition for adhesion sites and nutrients, secretion of antimicrobial compounds, reduction in inflammatory cytokines, improvement of intestinal barrier integrity, and production of butyrate and SCFA [85,86,87]. A similar effect could be exerted by GILZp, since we recently demonstrated that GILZp can restore an optimal environment for the colonization of health-promoting bacteria [9]. 

Our analysis of the fungal community revealed that colitic mice were characterized by reduced alpha diversity and by a distinct taxonomic composition in comparison to non-colitic mice. This is in line with previous findings [21,30,64]. Py and GILZp, when administered alone and in combination, were effective in reducing DSS-induced fungal dysbiosis and promoting a fungal composition that was similar to that of non-colitic mice. Furthermore, consistent with previous studies [64,88,89,90,91], our results indicate that the composition of the intestinal mycobiome in both non-colitic and colitic mice is dominated by fungi from the Ascomycota and Basidiomycota phyla. In particular, Ascomycota consistently appeared to be the most abundant phylum in all of the groups. In line with RT-qPCR results, an in-depth analysis of the mycobiota taxonomic composition showed that the DSS group was dominated by the *Meyerozyma* genus (phylum Ascomycota, family *Debariomycetaceae*, species *M. guillermondii* species, formerly *Candida guillermondii*). In accordance with Ming Li et al. [37], this result suggests that this genus could be one of the main microbial taxa involved in the pathogenesis of ulcerative colitis. Both GILZp and Py, when administered individually or in combination, resulted in a significant reduction in *Meyerozyma* abundance compared to the control group. Although the mechanistic basis for this effect requires further study, our data suggest that a reduction in gut colonization by *Meyerozyma* spp. could contribute to a positive outcome in patients with ulcerative colitis. Furthermore, our results showed that the *Mycosphaerella* genus was one of the predominant fungal taxonomic units in the intestinal fungal community of non-colitic mice (specifically *M. tassiana*, previously known as *Davidiella tassiana*, the sexual form of *Cladosporium herbarum*, which belongs to phylum Ascomycota, family *Mycosphaerellaceae*), whereas it was not significantly associated with colitic mice. Interestingly, this fungus showed a significant association with both the Py and Py + GILZp groups. Thus, there might be a potential link between *Mycosphaerella* and anti-colitic effects mediated by Py. Understanding the role of specific fungal species, such as *M. guillermondii* and *M. tassiana*, in restoring mycobiome balance is crucial for developing potential therapeutic strategies to promote gut health and alleviate gastrointestinal disorders.

## 5. Conclusions

Collectively, our data provide compelling evidence that GILZp and Py treatments, by enhancing intestinal barrier function and reducing inflammation and fungal dysbiosis, exert a protective effect in DSS-induced colitis. These findings represent a promising starting point for further investigation into the potential use of their combination as a novel therapeutic strategy for managing IBD.

## Figures and Tables

**Figure 1 biomolecules-14-01321-f001:**
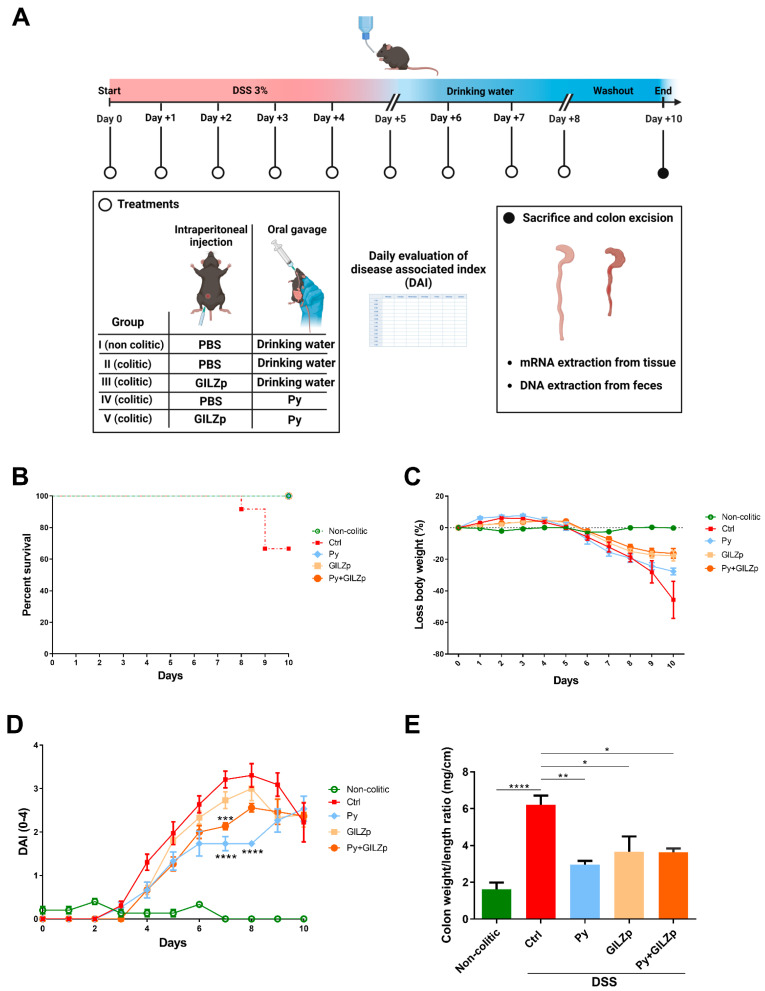
Ameliorative effect of individual and combined GILZp and Py treatments on DSS-induced colitis symptoms in mice. (**A**) Schematic diagram of the experimental design and procedures. Colitis was induced with the administration of 3% DSS for 5 days. From day 0 to day 8, the four colitic groups received GILZp (0.2 mg/kg), Py (1000 mg/kg), a Py + GILZp combination, or PBS. The schematic was created with Biorender BioRender (HomeStars, Toronto, ON, Canada)). (**B**) Kaplan–Meier curve depicting mortality in mice with DSS-induced colitis. (**C**,**D**) Body weight loss and clinical score (DAI) were registered daily over the whole experimental period. (**E**) Mean weight/length ratio values of colons. Values are expressed as the means ± SEM value from two independent experiments (*n* = 5–7 for each group). * *p* < 0.05, ** *p* < 0.01, *** *p* < 0.001, **** *p* < 0.0001. Where not indicated, a non-significant difference was observed.

**Figure 2 biomolecules-14-01321-f002:**
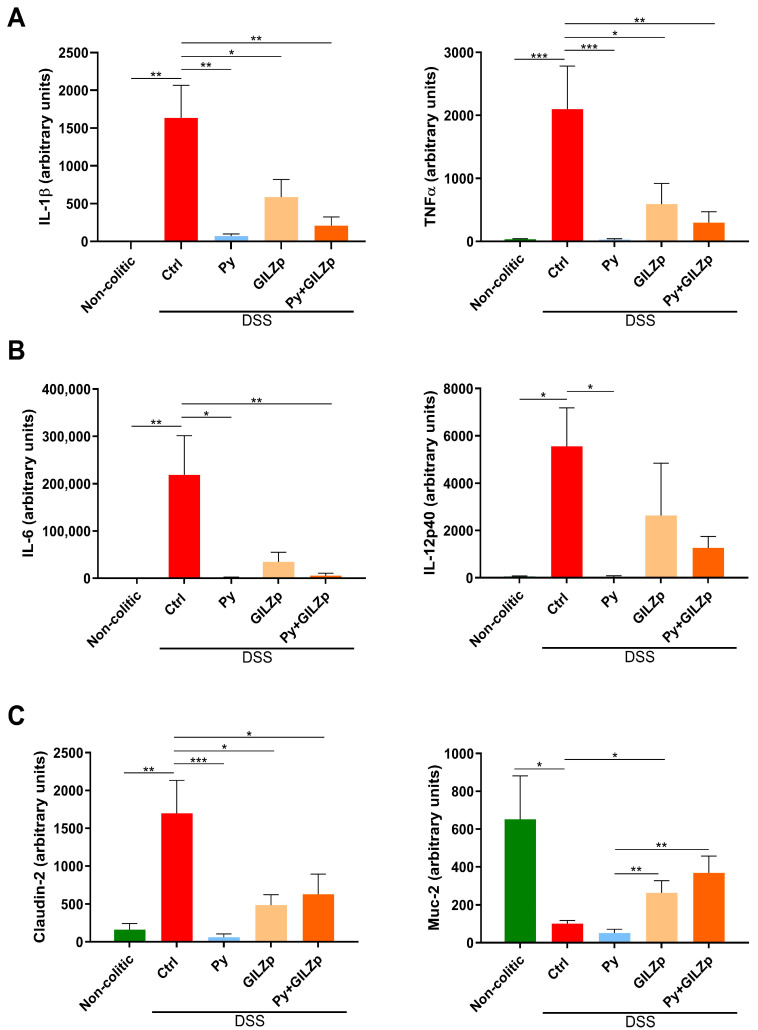
Suppression of the expression of inflammatory cytokines and restoration of intestinal barrier integrity with individual and combined GILZp and Py treatments. Quantitative RT–PCR analysis of IL-1β and TNF-α (**A**) expression, (**B**) IL-6 and IL-12p40 expression, and (**C**) Claudin-2 and MUC2 expression in the colons of non-colitic and colitic mice. The colitic mice were treated as indicated. Values are expressed as means ± SEM. (*n* = 5–7) for each group. * *p* < 0.05, ** *p* < 0.01, *** *p* < 0.001.

**Figure 3 biomolecules-14-01321-f003:**
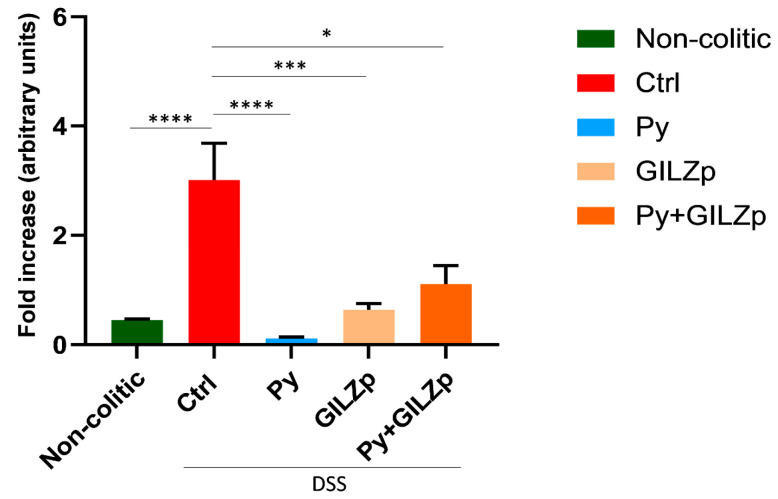
Restoration of *Candida* and former *Candida* species burden to physiological levels in treated colitic mice. Quantitative RT-PCR for the detection of *Candida* spp. burden in stool samples using a custom TaqMan probe and the ITS1-2 primers to amplify fungal rDNA. Differences between the cycle threshold (Ct) of *Candida* spp. and ITS1-2 were calculated (ΔCts), and data are shown as 2^−ΔCts^. Values are expressed as mean ± SEM in panel B (*n* = 4 per group). * *p* < 0.05, *** *p* < 0.001, **** *p* < 0.0001.

**Figure 4 biomolecules-14-01321-f004:**
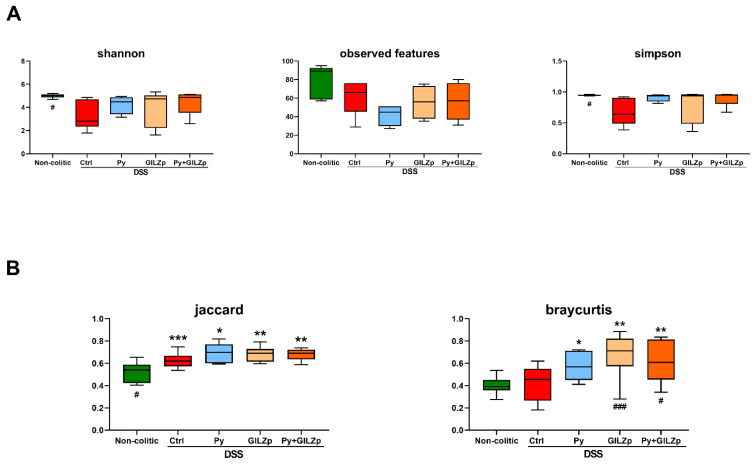
Diversity analyses showing the peculiar fungal composition of non-colitic, control, and treated mice. (**A**) Boxplots of alpha diversity indexes (Shannon, Observed Features and Simpson) of fecal mycobiota evaluated for data rarefied at 4000 reads. Statistical significances were determined using the Wilcoxon multiple-comparison test with FDR adjustment. ^#^ *p* < 0.05 for non-colitic vs Crtl. group. (**B**) Boxplots of the beta diversity distances (Jaccard and Bray–Curtis) of fecal mycobiota evaluated for data rarefied at 4000 reads. Statistical significances were determined using the Wilcoxon multiple-comparison test with FDR adjustment. The asterisks (*) indicate the significance of differences for the Crtl or treated group versus the non-colitic group, and the hashtags (^#^) indicate the significance of differences for the non-colitic or treated versus the Crtl group. ^#,^* *p* < 0.05, ** *p* < 0.01, ***^,###^ *p* < 0.001. (**C**) PCoA of Jaccard and Bray–Curtis distances. The first two PCoA components are represented in a scatter plot and in the marginal boxplots. The percentage of variance explained by each component is shown on each axis. Confidence ellipses assume a multivariate t-distribution. Asterisks (*) indicate the significance of differences for the control or treated versus non-colitic groups, and hashtags (#) indicate the significance of differences for the non-colitic or treated versus Crtl groups. ^#,^* *p* < 0.05, **^,##^
*p* < 0.01.

**Figure 5 biomolecules-14-01321-f005:**
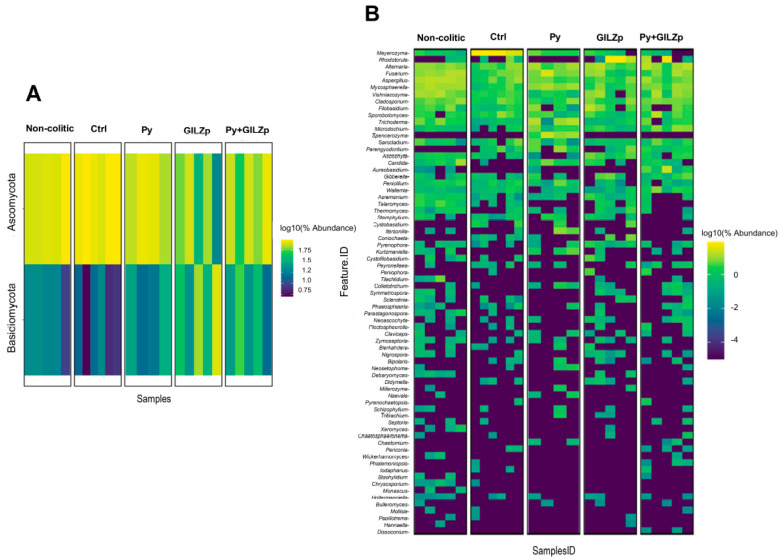
Mycobiome composition (phylum and genus levels) and mycobiome core (genus levels) analyses. (**A**) Heatmap depicting the abundances of phyla based on the sequencing data according to groups of interest. Features are sorted from top to bottom by the decreasing value in the average abundance calculated over the entire sequencing. (**B**) Heatmap depicting the abundances of genera present based on the sequencing data according to groups of interest. Features are sorted from top to bottom by decreasing value of the average abundance calculated over the entire sequencing. (**C**) Heatmaps of the mycobiome core of the genera in each sample group, as indicated in each heatmap, versus the per-sample relative abundance threshold (minimum detection threshold = 1%). The core of each group is composed of genera with a relative abundance of at least 1% and with prevalence greater than 50% in the group itself.

**Figure 6 biomolecules-14-01321-f006:**
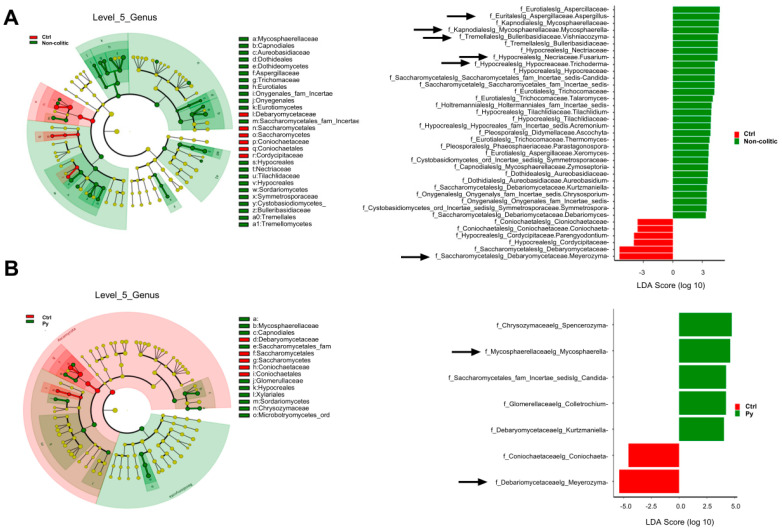
Differential association analysis of genera in each treated or non-colitic group in comparison with the control group. (**A**–**D**) Cladograms and histograms showing taxonomies, up to the genus level, that were significantly (*p* < 0.05) associated with one of the two groups considered, as indicated in the corresponding color legend. The yellow circles in the cladogram indicate non-significant taxa. The black arrows in the histograms emphasize the genera belonging to the mycobiome core of the group itself, from among the genera significantly associated with each group.

**Table 1 biomolecules-14-01321-t001:** DAI score.

Bleeding	Stool Consistency	Weight Loss (WL, %)	Value Assigned According to WL
1: presence of blood	1: moderate soft stools	1–5	1
2: moderate bleeding	2: soft stools	6–10	2
3: moderately high bleeding	3: soft stools and diarrhea	11–20	3
4: abundant bleeding	4: diarrhea	>20	4
1: presence of blood	1: moderate soft stools	1–5	1

**Table 2 biomolecules-14-01321-t002:** NuCel^®^ 582 MG specifications.

Dry Matter	Min.	94	g/100 g
Total nitrogen	Min.	10.2	g/100 g
Amino nitrogen	Min.	3.2	g/100 g
Proteins	Min.	63.8	g/100 g
pH		5.5–5.9	

**Table 3 biomolecules-14-01321-t003:** Amino acid and vitamin composition of NuCel^®^ 582 MG.

**Total Amino Acids (Average Value of Total Amino Acid Expressed on Product as Is)**					
Alanine (Ala)	4.7	g % g	LYSINE (LYS)	4.7	g % g
Arginine (Arg)	3	g % g	METHIONINE (Met)	0.2	g % g
Aspartic Acid (Asp)	5.6	g % g	PHENYL ALANINE (Phe)	2.4	g % g
Cystine (Cys)	1.2	g % g	PROLINE (Pro)	1.9	g % g
Glutamic Acid (Glu)	11.7	g % g	SERINE (SER)	2.8	g % g
Glycine (Gly)	3	g % g	THREONINE (Thr)	3.2	g % g
Histidine (His)	1.1	g % g	TRYPTOPHAN (Trp)	0.1	g % g
Isoleucine (Ile)	2.4	g % g	TYROSIN (Tyr)	1.3	g % g
Leucine (Leu)	3.9	g % g	VALINE (Val)	3	g % g
**Vitamins**					
Thiamine (B1)	Max. 50	ppm	Pyridine (B6)	Max. 50	ppm
Riboflavin (B2)	Max. 50	ppm	Biotin (B8)	Max. 50	ppm
PP—Niacin (B3)	Max. 500	ppm	Folates (B9)	Max. 50	ppm
Pantothenic (B5)	Max. 100	ppm	Cobalamin (B12)	Max. 200	ppm

## Data Availability

The metagenomics datasets generated during the current study have been deposited in the Sequence Read Archive (SRA) under BioProject ID number PRJNA1021184 (https://www.ncbi.nlm.nih.gov/bioproject/PRJNA1021184). All of the others generated during the current study are available upon request.

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
