# Peer review of "Glucocorticoid-Induced Leucine Zipper Protein and Yeast-Extracted Compound Alleviate Colitis and Reduce Fungal Dysbiosis"

_biomolecules, 2024, doi:10.3390/biom14101321_

Round 1

Reviewer 1 Report

Comments and Suggestions for Authors

I read this paper with interest. The authors examined a protein (GILZ) and a fungal extract (Py), both individually and in combination, in a model of experimental colitis, focusing on pro-inflammatory cytokines and epithelial barrier integrity proteins. The results suggest a combined use of these substances in experimental colitis to reduce the inflammatory burden.

Some minor comments:

1) Introduction: There is a complete lack of reference to other fungal extracts used in the treatment of experimental colitis models. Another recent ex vivo human colitis study used Hericium erinaceus derivatives (https://pubmed.ncbi.nlm.nih.gov/37465689/), which I believe the authors should cite, discuss, and compare with their results. Recent reviews discuss the benefits of this fungus in IBD, and there is also a study (i.e., HERICIUM-UC) on the potential real-life effects of fungal supplementation of this compound in patients with UC.

2) I would prefer the authors to use terms such as "DSS-induced colitis" or "experimental colitis" rather than "UC" when referring to this colitis. It is well known that DSS-induced colitis, primarily used for Th1/Th2 cytokines, mimics not only the pathogenic features of UC but also Crohn's disease.

3) Paragraph 2.2 should probably be expanded to ensure data reproducibility.

Author Response

Dear Editor,

We are grateful for the attention you have afforded to our manuscript and the consideration of our work by the reviewers and their constructive comments. We carefully considered the reviewers’ comments and suggestions. As detailed below, we modified the manuscript to address their concerns. We also edited the figures 5 and 6 as requested.

Replies to the reviewers follow:

Reviewer 1

1) Introduction: There is a complete lack of reference to other fungal extracts used in the treatment of experimental colitis models. Another recent ex vivo human colitis study used Hericium erinaceus derivatives (https://pubmed.ncbi.nlm.nih.gov/37465689/), which I believe the authors should cite, discuss, and compare with their results. Recent reviews discuss the benefits of this fungus in IBD, and there is also a study (i.e., HERICIUM-UC) on the potential real-life effects of fungal supplementation of this compound in patients with UC.

Reply: we thank the reviewer for this important omitted part. We added text on fungi benefits in the Introduction, page 4 and 5, lanes 86-96, adding references #45-53, and in the Discussion section, page 19, lanes 428-431.

2) I would prefer the authors to use terms such as "DSS-induced colitis" or "experimental colitis" rather than "UC" when referring to this colitis. It is well known that DSS-induced colitis, primarily used for Th1/Th2 cytokines, mimics not only the pathogenic features of UC but also Crohn's disease.

Reviewer 2 Report

Comments and Suggestions for Authors

This is a well-written manuscript that demonstrates Py and GILZ can improve the prognosis of colitis by inhibiting the expression of pro-inflammatory cytokines and Claudin-2, and both Py and GILZ could help restore intestinal burden. The results are well-organized and clearly described. I do recommend this manuscript for publication.

Minor: 

1) Please briefly describe the criteria for clinical score (DAI) in the method section. 

2) The figure quality/resolution of Fig. 5 and Fig. 6 is low. All labels are not shown correctly. Please carefully revise them. 

Author Response

Dear Editor,

We are grateful for the attention you have afforded to our manuscript and the consideration of our work by the reviewers and their constructive comments. We carefully considered the reviewers’ comments and suggestions. As detailed below, we modified the manuscript to address their concerns. We also edited the figures 5 and 6 as requested.

1) Please briefly describe the criteria for clinical score (DAI) in the method section.

Reply: we added a table with the detailed score system (page 6) 

2) The figure quality/resolution of Fig. 5 and Fig. 6 is low. All labels are not shown correctly. Please carefully revise them. We also improved the quality of Figure 4.

Reply: we agree about the low quality of these figures. Both figures and labels were modified and ameliorated.

Reviewer 3 Report

Comments and Suggestions for Authors

The manuscript by Gentili et al. examines the therapeutic effects of a recombinant GILZ protein and a yeast extract compound (Py) in a mouse model of colitis. 

The authors have shown previously that GILZ protein is associated with the gut barrier functions and the immune cell homeostasis. The current study evaluated the synergistic effects of the GILZ protein and prebiotic Py. The study is well-designed and directly addresses the hypothesis. The results seem to show that there are significant effects with Py alone, but the combination therapy showed a synergistic effect with the GILZ treatment. What is the advantage of using GILZ treatment if Py is available? The mechanisms by which the combined therapy promotes the Meyerozyma genus in the gut may be worth looking into in the future studies. One comment was that the mycobiome figures are not clear so that it was difficult to follow the part of the results.  

Author Response

Dear Editor,

We are grateful for the attention you have afforded to our manuscript and the consideration of our work by the reviewers and their constructive comments. We carefully considered the reviewers’ comments and suggestions. As detailed below, we modified the manuscript to address their concerns. We also edited the figures 5 and 6 as requested.

The manuscript by Gentili et al. examines the therapeutic effects of a recombinant GILZ protein and a yeast extract compound (Py) in a mouse model of colitis. 

The authors have shown previously that GILZ protein is associated with the gut barrier functions and the immune cell homeostasis. The current study evaluated the synergistic effects of the GILZ protein and prebiotic Py. The study is well-designed and directly addresses the hypothesis. The results seem to show that there are significant effects with Py alone, but the combination therapy showed a synergistic effect with the GILZ treatment. What is the advantage of using GILZ treatment if Py is available? The mechanisms by which the combined therapy promotes the Meyerozyma genus in the gut may be worth looking into in the future studies. One comment was that the mycobiome figures are not clear so that it was difficult to follow the part of the results.  

Reply: the advantage of using GILZ treatment is that no prebiotic, so far, can be considered a “pharmacological treatment” but as a “supportive” therapy for colitis. GILZ, as a protein, can represent a possible pharmacological tool to be flanked by prebiotics, which has never been explored before. The only PY, even though has been demonstrated to be effective in our study, might not be sufficient for a durable therapy. Furthermore, our study is the first to demonstrate that the promotion of the Meyerozyma genus is associated with the restoration of intestinal barrier integrity and a reduction in pro-inflammatory cytokines in an experimental model of IBD. A deeper understanding of the mechanisms driving this effect could undoubtedly offer valuable insights into the therapeutic potential of modulating the gut mycobiome for treating inflammatory bowel diseases.